# Flower Colour Polymorphism, Pollination Modes, Breeding System and Gene Flow in *Anemone coronaria*

**DOI:** 10.3390/plants9030397

**Published:** 2020-03-23

**Authors:** Amots Dafni, Hagai Tzohari, Rachel Ben-Shlomo, Nicolas J. Vereecken, Gidi Ne’eman

**Affiliations:** 1Department of Evolutionary Biology and Institute of Evolution, University of Haifa, Haifa 3498838, Israel; hagai11@gmail.com; 2Department of Biology and Environment University of Haifa-Oranim, Tivon 36006, Israel; ekly@research.haifa.ac.il; 3Agroecology & Pollination Group, Landscape Ecology & Plant Production Systems, Université Libre de Bruxelles (ULB), Boulevard du Triomphe CP 264/2, B-1050 Brussels, Belgium; Nicolas.vereecken@ulb.ac.be

**Keywords:** *Anemone coronaria* L., flower colour polymorphism, bees, flies, pollination, population genetics

## Abstract

The flower colour of *Anemone coronaria* (Ranunculaceae) is a genetically inherited trait. Such intra-specific flower colour polymorphism might be driven by pollinators, other non-pollinating agents, or by abiotic factors. We investigated the genetic relations among red, white and purple-blue flower colour morphs growing in 10 populations of *A. coronaria* in Israel, in relation to their breeding system, pollination modes, differential perception by bees and visitors’ behaviour. Flowers of these three morphs differed in their reflectance that could be perceived by bees. Honeybees, solitary bees and flies demonstrated only partial preferences for the different colour morphs. No spontaneous self-pollination was found; however, fruit set under nets, excluding insects but allowing wind pollination, was not significantly lower than that of natural free pollinated flowers, indicating a potential role of wind pollination. *Anemone coronaria* flowers were visited by various insects, honeybees and *Andrena* sp. preferred the white and purple-blue morphs, while the syrphid flies preferred the white flowers. Thus, visitor behaviour can only partially explain the evolution or maintenance of the colour polymorphism. No significant genetic differences were found among the populations or colour morphs. Wind pollination, causing random gene flow, may explain why no significant genetic divergence was found among all studied populations and their colour morphs. The existence of monomorphic red populations, along other polymorphic populations, might be explained by linked resistance to aridity and/or grazing.

## 1. Introduction

Flower colour polymorphism (hereafter FCP) refers to the variation in flower colours within or between natural populations of the same species. FCP may include cases of gradual [1], but mainly refers to discrete differences in flower colours among morphs [2,3,4,5,6]. FCP may occur sympatrically [7], parapatrically [8], or allopatrically [9].

Pollinators’ constancy, especially of bees, occurs when they return to feed on any preferred rewarding flower after quickly learning its visual and chemical properties. Such constancy to one flower morph and avoidance of others may cause divergent evolution [10]. Therefore, the coexistence of multiple colour morphs within any species requires special attention. FCP may be explained by pollinator mediated selection (e.g., [11,12]), antagonist selection by pollinators and herbivores (e.g., [11,12,13]), or adaptations to abiotic factors that are linked to inherited floral traits [14,15].

Pollinators may maintain FCP when colour preferences vary among pollinator guilds, morphs, individuals, foraging bouts, or when it varies with the frequencies of FCP [2,11,16,17,18,19,20,21]. A number of studies have shown that balancing and/or negative frequency-dependent selection may explain FCP [20,22,23]. Pollinators may also have fluctuating preferences depending on nectar properties, plant or inflorescence height, warmth, or flower abundance [11,16,21,22,24,25,26,27].

In some cases, the strength of the pollinators’ preferences may be insufficient to affect the maintenance of FCP [28,29].

When pollinators discriminate among FCPs, gene flow is expected to be more frequent within each colour morph than between morphs [7]. Consequently, genetic variation is expected to be lower within morphs than between them [30].

Flower traits act as signals not just for pollinators but also for herbivores that can discriminate among flower colour morphs [31,32,33], and their feeding preferences may vary among sites, over time, or across taxa [14,34,35]. Consequently, the same flower trait may make plants conspicuous to both pollinators and herbivores that may drive different selection pressures [34,36,37,38].

When flower colour morphs that are preferred by pollinators are also more susceptible to herbivores, there might be a conflicting selection pressure on flower colour [39]. Flower colour may have pleiotropic effects on plant defence [40].

Correlations between petal colour and herbivore defence chemicals in leaves or fruit have been observed [31,41] and may affect FCP either via genetic linkage or pleiotropy [39,42,43]. Such correlations may contribute to indirect selection by herbivores [31,32,33]. Herbivores that oviposit in flowers, causing pre-dispersal seed predation, are likely to use the same flower cues as pollinators to locate their host flowers [43]. Pollinators and predators may ultimately exert conflicting pressures on FCP because flowers that are most visited by pollinators are likely to produce more food for pre-dispersal seed predators [12,39], but see [13].

Selection along abiotic gradients may also maintain FCP, in part because anthocyanins are linked to increased tolerance to extreme drought, heat, pests, soil infertility, or to UV radiation [28,29,44,45]; but anthocyanin production also has costs [25,26,28,29,46].

Experiments showed selection on flower colour due to correlations between petal colour and heat tolerance [29,42,47]. Fluctuating selection associated with rainfall contributes to the maintenance of a blue-white flower colour polymorphism in Linanthus parraye [4,5], but anthocyanin production has also cost [25,26,28,29,46].

*Anemone coronaria* L. is an east Mediterranean species [48]. growing in diverse habitats around the Mediterranean Sea, and each plant produces 1–10 flowers, depending on age and local conditions. It exhibits an extreme case of FCP with red, white and purple-blue discrete colours, and with at least 30 gradual colour variants (Figure 1) [48,49]. Flower colour is determined by the anthocyanins and flavonoids content of the sepals [49].

At least four epistatic genes that act in the following sequence, Alb (albino), Sc (scarlet), W (white) and B (blue), were found to be responsible for flower colour [50,51,52]. Red *A. coronaria* populations were found to be pollinated mainly by Glaphyrid beetles [53] but also by honeybees [54]. Glaphyrid beetles find shelter and mate within large red flowers of *A. coronaria*; some floral toxic chemicals and the red background provide them some protection from bird predators [55,56].

Populations growing in semi-arid areas in southern Israel are monomorphic red. Many other populations have various mixed colours, while a few northern ones are purely white. Consequently, it has been hypothesised that the red morph is connected to aridity, while the white and purple-blue morphs are connected to rock and soil properties [50]. Red genotypes of *A. coronaria* were found to be resistant to heavy cattle grazing, and their population declined when grazing stopped [57], but not to roe-deer grazing [58]. Our aims were: (1) To explore the role of FCP in the pollination ecology of *A. coronaria*. (2) To examine the degree of genetic differences among populations and among flower morphs within populations. (3) To propose possible mechanisms for the maintenance of FCP in *A. coronaria* in Israel.

To do so, we asked the following questions: (1) Are different colour morphs (as seen by humans) also perceived differently by bees? (2) Are different colour morphs favoured by different flower visitors? (3) What are the possible roles of spontaneous self and wind pollination? (4) Does the genetic structure indicate the possibility of reproductive isolation among populations or between colour morphs within populations?

## 2. Results

### 2.1. Flower Reflectance and Perception

The different flower colour morphs (red, white and purple-blue) of *A. coronaria* L. (Megiddo site) differ markedly in their relative reflectance (Figure 2), and in their locations in the bee colour perception hexagon (Figure 3). The average (± *SD*) location of the red flowers (*N* = 20) was X = −0.08850 ± 0.08869 and Y = −0.03450 ± 0.01050, that of the white flowers (*N* = 20) was X = 0.08450 ± 0.03069 and Y = 0.14350 ± 0.02207, and that of the purple-blue flowers (*N* = 20) was X = −0.1730 ± 0.04438 and Y = 0.1675 ± 0.03024. The pairwise average Euclidean distances between the average locations of the different colour morphs were: purple-blue to red: 0.261 ± 0.050, purple-blue to white: 0.287 ± 0.050, red to white: 0.269 ± 0.079.

### 2.2. Number of Carpels

The average (± *SD*) number of ovules in red (*N* = 5), purple-blue (*N* = 5) and white (*N* = 5) flower morphs, from the Megiddo population, was 584.6 ± 75.7, 582.1 ± 69.1 and 573.8 ± 50.3, respectively, with no significant differences among the colour morphs (*F*_2,14_ = 0.007, *P* = 0.993). 

### 2.3. Pollination Modes

#### 2.3.1. Spontaneous Self-Pollination

Under experimental exclusion of both pollinator activity and of wind pollination, in the Beit Lehem Haglilit site, there was no fruit set in red, white and purple-blue flower morphs (*N* = 20 flowers for each colour morph), indicating that there was no spontaneous self-pollination.

#### 2.3.2. Insect and Wind Pollination

In the 2012 season, the average (± *SD*) percentage of fruit set in the pollinator exclusion cages was 78% ± 10% (*N* = 11). In the 2014 season, the percentage of fruit set under natural free pollination (insects and wind) and of only wind pollinated flowers (under nets) in all colour morphs, ranged from 95.8% to 100% (*N* = 18, six from each colour morph) and from 90.3% to 96.1% (*N* = 18, six from each colour morph). There were no significant differences in fruit set among the colour morphs under free pollination (Kruskal–Wallis, χ = 1.11, df = 2, *P* = 0.775), and no significant differences among the colour morphs under pollinator exclusion nets (Kruskal–Wallis, χ = 1.72, df = 2, *P* = 0.632). However, there was a small but significant difference in the average percentage (± SD) of fruit set between open (98.0 ± 1.2%) and net covered (93.0 ± 1.7%) plants (all colours pooled (Mann–Whitney *U* = 160, *Z* = 2,519, *P* = 0.012, *N* = 24).

During three periods of 24 hours, 30 airborne pollen traps captured on average (± *SD*) 36.3 ± 0.6 (*N* = 80) *A. coronaria* pollen grains. Considering trap area and duration, the average pollen deposition rate was 0.05 grains per hour per mm^2^. The average (± *SD*) stigma surface (measured under microscope) was 0.491 ± 0.0092 mm^2^ (*N* = 100), and the average duration of stigmatic receptivity of a flower is 3.5 days (= 84 h) [54]. Thus, during a flower’s lifetime, 2.1 pollen grains can be deposited by wind on each stigma that contains a single ovule.

We interpreted any fruit production under pollinator exclusion as a result of wind pollination due to: (1) the absence of autogamy in *A. coronaria* (see the results), (2) the spatial separation of individual flowers under the cages (about 5 cm between adjacent flowers), and (3) flower morphology, which decrease the probability of pollination due to contact between neighbouring flowers is expected to be low.

### 2.4. Flower Visitors

During 2011 and 2012, the flower visitors were mostly honeybees, solitary bees and flies, while almost no beetles were observed. The total number of visitors that landed on flowers of all colour morphs was extremely low (175), and the average (± *SD*) visitation rate was only 0.072 ± 0.078 visits per flower per hour. Due to the scarcity of visitors, this dataset was insufficient for analysing whether or not there was a preference of any visitor type to any flower colour morph.

Four independent observation sessions were made on 6.02.2014 (two sessions), 9.02.2014 and 21.02.2014, in which the sequence of the visits of 13, 120, 30 and 43 individual bees to the various colour morphs were recorded, respectively (Table 1). Honeybees, solitary bees (mostly *Andrena* sp.), and syrphid flies mostly visited the white and purple-blue flower morphs, usually avoiding the red ones. During that season, only one beetle was observed visiting a red flower (Table 1). A χ^2^ test revealed that honeybees and *Andrena* sp. did not visit the colour morphs randomly, according to their frequencies, by avoiding visiting the red flowers (Table 1).

### 2.5. Genetic Structure of Populations and Colour Morphs

The amplified fragment length polymorphism (AFLP) analyses revealed 462 polymorphic loci. The levels of polymorphism (*P*) and gene diversity (*He*-unbiased expected heterozygosity) for each population and colour morph are summarized in Table 2. The mean (± *SE*) polymorphism level was 70.67% ± 5.25%, and gene diversity level was 0.124 ± 0.002 (Table 2). Nei unbiased genetic identity (*I*) between the tested populations (1. Mt. Meron, 2. Zippori, 3. Mt. Carmel (University), 4. Mt. Carmel (Nof Carmel), 5. Alonim, 6. Beit Shearim, 7. Balfouria, 8. Megiddo, 9. Purra, 10. Lahav) was very high (*I* = 0.99), indicating no differentiation between either populations or colour morphs (Exact test, *P* > 0.99).

Analysis of molecular variance (AMOVA) [60] revealed that the major part of the molecular variance (99%) was found within populations (Φ_9_,_226_ = 0.014, *P* = 0.001). Likewise, the Bayesian assignment test [61,62], indicated no differentiation between populations or colour morphs. The highest delta-K value (where K is a putative number of populations) of 33.8 was found for K = 2, and further for K = 4 (32.6), where the structure bar-plot (representing all tested individuals) appeared to be similar through the sampling range (Figure 4).

## 3. Discussion

### 3.1. Colour Discrimination by Visiting Insects

Colour morphs of the *Anemone coronaria* flower (Figure 1) clearly differ in their relative reflectance curves (Figure 2) and to the human eye. However, because different insects differ in the sensitivity of their photoreceptors to various spectral wavelengths, colour vision perceptual models must be used to examine whether or not an insect can distinguish between any given colours [59]. The threshold values of Euclidean distance between any two colours in the hexagon units above which bees can perceive them as different signals ranges between 0.062 [63,64] and 0.100 [65]. The distances between the three *A. coronaria* colour morphs ranged between 0.26 and 0.27. Therefore, it can be assumed that honeybees, and probably also solitary bees, can differentiate among them. The centre of the hexagon represents the background colour, thus the shorter distance of the red flowers from this point indicates that bees can differentiate them from the background less than white or purple blue flowers.

Direct evidence for colour vision, the ability to discriminate colours according to spectral shape but independent of intensity, has been demonstrated for only a few fly species [66]. Therefore, we do not really know to what extent hoverflies (Syrphidae), which were observed visiting *A. coronaria* flowers, can discriminate between its colour morphs.

Although almost not recorded in our study, the interaction between the beetles of the Glaphyridae family and the red bowl-shaped flowers in the Mediterranean region is well-documented [53]. These beetles, in contrast to bees, have photoreceptors that are sensitive to the UV, green and red ranges of the spectrum [67].

Thus, it seems that the combination of different colour reflection curves among *A. coronaria* colour morphs, and the different sensitivity of the photoreceptors of the major flower visitors (bees, flies and Glaphyrid beetles) could have been the basis for differential visitation, and pollination of any given flower colour morph by any visitor or pollinator type.

### 3.2. Spontaneous Self-Pollination and Wind Pollination

Earlier field experiments demonstrated that *A. coronaria* is strongly self-incompatible, and hybridization experiments showed that the different colour morphs could be readily crossed as they produced achenes with viable seeds [50].

Our results show the absence of spontaneous self-pollination in all flower colour morphs of *A. coronaria*. In contrast, a very high rate of fruit production (98%) under natural free pollination and only slightly less (93%) under insect exclusion with a net cover that allowed wind pollination. These surprising results indicate the possibility that wind pollination might cause free gene flow among populations and between colour morphs within them, independent of possible pollinators’ preferences to any flower colour morph. Since *A. coronaria* is a widespread species with almost continuous dispersal, wind pollination may contribute to free gene exchange and thus mask the role of the various putative agents, which may maintain colour polymorphism.

Ambophily is a pollination syndrome by both wind and insects and is generally related to harsh environmental conditions with low activity of pollinators [68,69,70,71]. Ambophily has evolved independently at least three times in the Ranunculaceae, in *Aconitum gymnandrum* [69], in *Ranunculus weyleri* [72] and in *Thalictrum* [73]. The Mediterranean winter is mild, and thus it is hard to explain the presence of ambophily in *A. coronaria* solely on environmental selective pressure via the scarcity of pollinators. Therefore, we propose it is a kind of ’pollination assurance’ mechanism, under unfavourable rainy conditions [74]. *Anemone coronaria* flowers close at night and on rainy days, thus its pollen is protected and wind pollination can occur also during sunny days, which are not uncommon in the mild Mediterranean winter.

### 3.3. Flowers Visitors

The red morph of *A. coronaria* was studied previously, and Glaphyrid beetles were found to be its major pollinators [53]. In contrast to bees, these beetles have red receptors [67]. However, early in the season, when other flower resources are scarce, red *A. coronaria* flowers are also visited by honeybees [54]. In the present study, honeybees were the most frequent visitors to *A. coronaria* flowers visitors, followed by solitary bees (mostly *Andrena* sp.) and syrphid flies, and only one visit of a Glaphyrid beetle was observed (Table 1).

Honeybees and *Andrena* sp. did not visit the colour morphs randomly, according to their frequencies, by avoiding visiting the red flowers (Table 1). The differences in the composition and frequency of flower visitors to *A. coronaria* flowers, in our study and among other studies [54,55,56], may reflect the high inter annual variability in temperatures and precipitation during the Mediterranean winter.

### 3.4. Population Genetic Analysis

Molecular analyses of ten *A. coronaria* populations consisting of red, white, and purple-blue colour morphs located along a rainfall gradient in Israel revealed high levels of polymorphism and genetic diversity (Table 2), and high genetic identity (similarity) among populations. The major part of the molecular variance was found to be within populations with no differentiation among populations or between colour morphs. In conclusion, the genetic analyses substantiate a free gene flow among all populations and colour morphs. These results raise questions concerning the evolutionary drivers and the ecological mechanisms that created and maintain this extreme case of flower colour polymorphism.

### 3.5. The Evolution and Maintenance of Flower Colour Polymorphism

The selective mechanisms that might create and/or maintain within-population variation in flower colours are diverse. For example, temporal or spatial variation in the strength and direction of natural selection on flower colour (i.e., fluctuating selection) can maintain genetic variation [75,76,77].

Flower colour is often linked with other plant traits that may directly affect plant fitness. Therefore, variation in flower colour could be maintained through selection on the linked traits rather than by direct selection on the colour itself [39,40,42,78,79]. Indirect responses to selection on correlated characteristics through non-pollinator agents may also influence the evolution of flower traits [43].

Pollinator-mediated selection is frequently proposed to explain polymorphism, with different colour morphs likely reflecting selection driven by different pollinators with different colour preferences [7,11,21,30,80]. If the pollinators are able to discriminate between colour morphs (e.g., *Iris lutescens* [2]), they could potentially exert selection driving flower colour polymorphism, depending on their behavioural responses as well as preferences to such colour differences (innate preference for particular colour morph, or learning process), as was shown for bees [12,81,82,83] and for birds [8].

Our results present clear evidence for specific preferences of flower colour morph by flower visitors. Honeybees and *Andrena* bees avoided visiting red flowers (Table 1), in contrast to earlier findings [54]; but they visited the white and purple-blue flowers, as expected from their frequencies. The differences in honeybees’ visitations may reflect differences in the environmental conditions as well as in the co-flowering species in the different studies. Thus, bees can only partially contribute to the isolation of the red colour morph by specific visitation and pollination. Syrphid flies visited the different flower colour morphs as expected by their frequencies and thus also cannot contribute to the isolation of any colour morph (Table 1). Glaphyrid beetles, which were almost absent from our study, were found in the past to be its major pollinators, visiting mainly red *A. coronaria* flowers [53]. Even in the absence of preference at the pollinator species level, the constancy of bees to any flower morph during a single bout may contribute to higher gene flow within than between morphs. Moreover, as pollinator preferences are related to flower frequency, their preferences may vary among locations, years and seasons.

*A. coronaria* starts flowering early in January, when pollinator activity is still very low [74]. During this time, the red flowering morph is abundant and visited by Glaphyrid beetles [53]. However, at the beginning of the flowering season, red flowers are visited also by honeybees as a result of their abundance and the scarcity of other pollen sources [54]. If the beetles were present only in the south of Israel, this could have explained the distribution of pure red populations there, but they are common all over the country, where mixed coloured populations occur (A. Dafni Pers. Obs.).

The geographic distribution of flower colour morphs of *A. coronaria* in Israel indicates that the southernmost populations consist of pure red genotypes. This distribution pattern has raised the hypothesis for pleiotropic relations between the red flower colour genotype and the resistance to aridity [48,84]. Indeed, the red flowering populations in the arid south also have narrower leaves in comparison to the same morph in the north (Y. Tankus, pers comm.). It is logical to assume that the smaller leaves confer resistance to aridity more than anthocyanin’s quantity [28,29,44,45]. Since the red morph is also common in north Israel, where it has larger leaves, it is inferred that leaf size exhibits a phenotypic plasticity and the red colour and leaf size seem not to be genetically linked. Moreover, to the best of our knowledge, beyond the above-mentioned correlation, there is no genetic evidence that supports this hypothesis.

The resistance of the red genotypes of *A. coronaria* to heavy cattle grazing [57] might be an advantage over other flower colours. However, since most open sunny areas, which are the typical habitats of *A. coronaria,* are under medium to heavy grazing pressure, this cannot explain why some populations are pure red and some have all colour morphs. Roe deer graze on *A. coronaria* flowers [58], but at present, it is not part of the native fauna of Israel.

The distribution of *A. coronaria* colour morph in Israel may also be related to soil type. Pure white and purple-blue colour morphs grow mainly on chalk-free soils on limestone or basalt rocks, respectively, and the red morph is dominant on relatively dry and calcareous soils [48,52,84]. However, this explanation seems non-satisfactory, since monomorphic red populations are also found growing close to other polymorphic populations in similar habitats, and more detailed research is needed to clarify this aspect.

We do not have any explanations why the monomorphic populations are almost exclusively red and why other populations are polymorphic. This pattern fits the generalization of Narbona et al. [6] who reviewed flower colour polymorphism in the Mediterranean (but omitted the case of *A. coronaria)* and concluded that, “The fact that no case with only polymorphic population was found suggests that flower colour in species with flower polymorphism is subjected to different evolutionary processes throughout their distribution”.

## 4. Materials and Methods

### 4.1. Study Sites

Flower visitor activity was observed in 2011, 2012 and 2014 in six mixed colour morph populations in northern Israel (Table 3, Figure 5). Colour reflection of different colour morph flowers was conducted in Megiddo site. We focused on the three major flower colour morphs (red, white and purple-blue). Plant material for DNA extraction was collected from 10 sites located throughout the main distribution areas of *A. coronaria* in Israel (Table 3). Each study site was about 1000 m^2^.

### 4.2. Flower Reflectance and Its Perception

To measure the relative (%) reflectance of *A. coronaria* we randomly selected 15 flowers of *A. coronaria*, each flower from a different individual growing in Megiddo site. We used a portable spectrophotometer (AVASPEC-2048-USB2-UA; Avantes, Eerbeek, The Netherlands) equipped with a Xenon light source (AVALIGHT-XE; Avantes) to measure the relative (%) reflectance (300–700 nm in 5 nm steps). The spectrophotometer was calibrated with a white standard (WS-2; Avantes). We measured the relative (%) reflectance of the centre of fresh petals, and we used the spectral sensitivity functions of the honeybee (*Apis mellifera*), because they are considered representative of a wide taxonomic spectrum of hymenopterans, as found in our study; furthermore, they are considered consistent within the Apoidea (bees sensu lato) [85,86,87].

We plotted the mean relative (%) reflectance of the petals across the 300–700 nm range. We then converted the raw data of relative reflectance measurements of petals into individual loci in the bee colour hexagon [59] by using the honeybee receptor-sensitivity curves and a green vegetation background measured in situ (Figure 3). We assessed and quantified the colour and achromatic contrasts between the petals of different colour morphs by calculating (a) pairwise Euclidean distances between loci and (b) the mean Euclidean distance between the species centroids in the bee colour hexagon. The Euclidean distance between any two loci indicates the perceived colour difference or contrast between the stimuli, and threshold values of hexagon units for colour discrimination usually range between 0.062 [63,64] and 0.100 [65] for bees.

### 4.3. Number of Carpels

The number of carpels in red (*N* = 5), purple-blue (*N* = 5) and white (*N* = 5) flower morphs, from the Megiddo population, were counted. Differences in carpel number among flower colour morphs were tested using ANOVA.

### 4.4. Pollination Modes

#### 4.4.1. Spontaneous Self-Pollination

To examine the possibility of spontaneous self-pollination (mechanical/autogamy), 60 random individuals, 20 from each colour morph (red, purple-blue and white) of flowers with flowering buds, growing in Beit Lehem Haglilit, were covered with fine nets (0.5 × 0.5 mm), which prevented the entry of insects and the movement of pollen by wind in our study. Flowers were covered on 25.1.10 and on 21.1.11 and were opened after 45 days to count fruit set.

#### 4.4.2. Wind Pollination

To examine the possibility of wind pollination, we constructed 30 pollen traps from plastic straps with 28.27 mm^2^ holes and sticky tape underneath them. The airborne pollen traps were mounted among the flowers for 24 h periods (16.01.11, 26.02.11 and 6.03.11) in the Megiddo population. Pollen grains of *A. coronaria* were identified by their size and shape, discriminated from others, counted, and deposition rate per unit area was calculated.

To estimate the rate of wind pollination, insects were excluded using cages (40 × 40 ×75 cm) covered with standard mosquito net (1 × 1 mm), which prevented insect entrance but allowed air currents. (We found no thrips, which may transfer pollen, at this study site). Eleven cages were set up in Megiddo during the 2012 season, each with an average (± *SD*) of 3.2 ± 1.95 flowers per cage (all colours combined). Thus, we repeated this experiment in Beit Lehem Haglilit in the 2014 season, with twelve randomly selected plants of each colour morph. Six plants of each colour morph were covered just before the opening of the first flower and their colour was marked. Six other plants were left uncovered as a control for free pollination. Plants were caged on 29.1.14, and the number of flowers that have set fruits (number of achenes was not counted) was recorded on 2.3.14.

The experiments were conducted in one plot of 100 × 200 m, presumably exposed to the similar wind regime.

### 4.5. Flower Visitors

Initial observations on visitors’ identity (their relative visit frequencies to each colour morph) were conducted in February and March 2011 and 2012, with additional observations in 2014. In total, we performed 18 observation sessions on average (± *SD*) 124 ± 45 min in six populations (Table 1) over a total of 37 h. In each population, we performed three observations, each on a separated date.

To check any possible colour preference by flower visitors we followed individual visitors. We recorded the colour morph of each flower visited from the first time the insect landed on the flower until it left the observation area (about 1000 m^2^). For each observation session, we also counted the number of each colour morph along five random 100 m long transects in the same area. These observations were conducted only on bright windless days, when the temperatures were above 20 °C. We used Pearson’s χ^2^ to determine whether there is a significant difference between the expected visit frequencies (frequencies of flower morphs) and the observed visit frequencies, as presented in Table 1.

### 4.6. Genetic Structure of Populations and Colour Morphs

Gene diversity and genetic identity among ten populations and between their colour morphs were tested using amplified fragment length polymorphism (AFLP) technique. The AFLP method was carried out essentially as described by Vos et al. [88]. High-quality genomic DNA (~200 ng) was digested with a pair of restriction enzymes (*Eco*RI/*Mse*I) at 37 °C for 4 h, then ligated to double stranded *Eco*RI (E -) and *Mse*I (M -) adaptors. The resulting fragments were amplified with non-selective primers, where the ligated adaptors served as target sites for primer annealing. Four selective primer combinations were used for AFLP amplification, E-TG/M-CAC, E-AG/M-CTT, E-TC/M-ACC and E-AC/M-CTG (E and M representing the restriction site and its ligated adaptor sequence). The selective *Eco*RI (E) primers were labelled with florescent dye (6-Fam, Ned, Vic and Pet, respectively). PCR reactions were carried out in a total volume of 13 L. PCR amplification cycles started at an annealing temperature of 65 °C, after which the annealing temperature was lowered by 0.7 °C per cycle for 12 cycles (a touch-down phase of 13 cycles), followed by 23 cycles at an annealing temperature of 56 °C. Amplification products were visualized under a Florescence-Reader (Applied Biosystems). Fragment analyses and genotyping were determined manually from the chromatographs using Peak-Scanner software (Applied Biosystems). DNA of several samples (~10%) were amplified and run in duplicate to validate the integrity of amplifications. The similarities between duplicated fingerprints were higher than 98%.

Amplification products were scored as discrete character states (present/absent) and transformed into band frequencies. Diversity values were based on phenotype frequency (phenotypes defined by the band patterns produced by individual primer pairs). Data were analysed by Tools for Population Genetic Analyses (TFPGA) software version 1.3 [89] and by GenAlEx 6.4 [60]. These programs consider AFLP bands as diploid-dominant markers in which the estimated allele frequencies are based on the square root of the frequency of the null (recessive) genotype. Genetic identity was calculated according to Nei [90], implemented in TFPGA [89]. Population differentiation was tested by exact tests (TFPGA [89]) 1000 dememorization steps, 10 batches, 2000 permutations per batch [91]. Molecular analysis of variance (AMOVA) was conducted using GenAlEx [60].

Bayesian partitioning clustering was applied using the program STRUCTURE 2.3.3 [61,62]. The data set was partitioned into 23 probable groups considering each colour morph in every population as a different group. The STRUCTURE run was applied, with four replicates for each putative group, 100,000 burn-in iterations and 100,000 Markov Chain Monte Carlo (MCMC) reps. The inference of the probable number of clusters was extracted by the log likelihood for each putative number of populations (K), Ln P(D) = L(K), and by the delta K method [92], using the program Structure Harvester [93].

As there is only one species of *Anemone* growing in Israel, we could not find a suitable closely related outgroup for our analyses. However, using the exact same method, in parallel to these analyses, we analysed several other species (e.g., *Pinus halepensis*, *Sternbergia clusiana*) that produced completely different AFLP profiles. The different AFLP patterns of these species, and the presence of variation in the AFLP profiles of *A. coronaria*, in the current study, can be regarded as controls for the present procedure.

## 5. Conclusions

Our results show that the extreme differences in flower colours among the human-perceived red, white and purple-blue morphs of *A. coronaria*, can also be sensed by bees, and earlier data showed that large red flowers are attractive to beetles. We revealed partial preferences of bees and flies to colour morphs. In other words, pollinators’ behaviour can only partially explain the evolution or maintenance of flower colour polymorphism in *A. coronaria*. We found no spontaneous self-pollination but fruit set under only wind pollination was only a little but significantly less than under free pollination. We also found no evidence for genetic divergence among populations and flower morphs indicating the absence of any reproductive barrier between them. Thus, *A. coronaria* in Israel has one panmictic meta-population with no indication of divergence among flower colour morphs. Free gene flow, due to effective wind pollination, explains the absence of genetic divergence among populations and flower morphs.

To the best of our knowledge *A. coronaria* is the first and only described case of a wind pollinated panmictic species with extreme flower polymorphism, but the adaptive value of the various colour morphs is still unclear.

## Figures and Tables

**Figure 1 plants-09-00397-f001:**
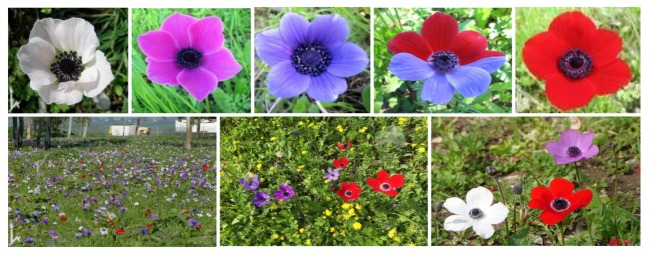
Red, white and purple-blue flower morphs of *Anemone coronaria* in Israel, and some multi-coloured populations.

**Figure 2 plants-09-00397-f002:**
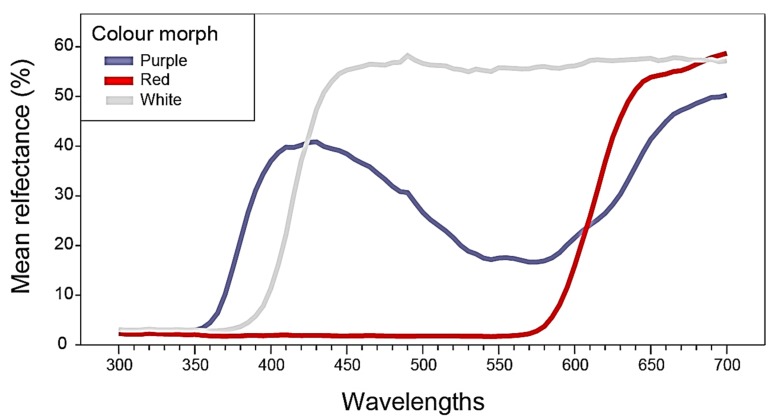
The relative reflectance (%) curves of red, white and purple-blue *A. coronaria* flower colour morphs.

**Figure 3 plants-09-00397-f003:**
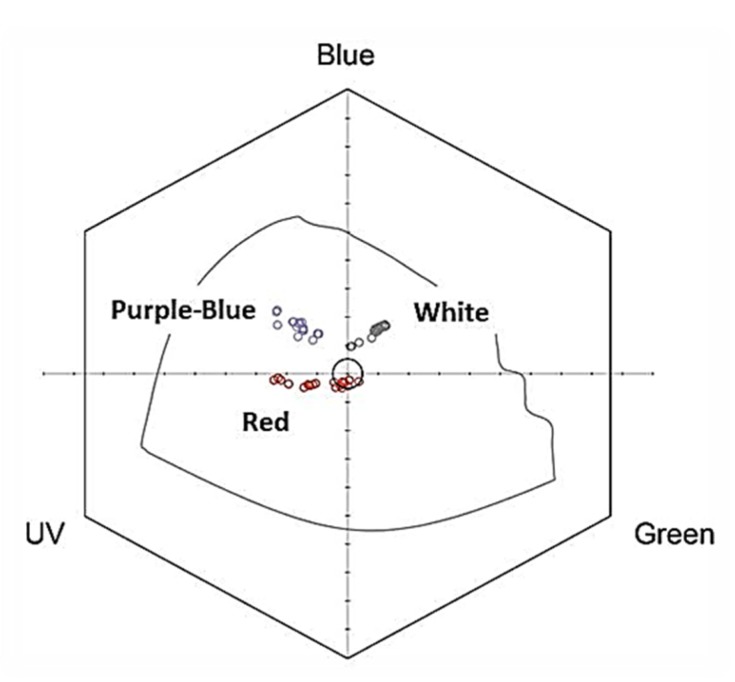
The perception of red, white and purple-blue flower morphs of *A. coronaria* by honeybees demonstrated by their location in the bee colour hexagon [59]. Axes origin represents green leaves background.

**Figure 4 plants-09-00397-f004:**
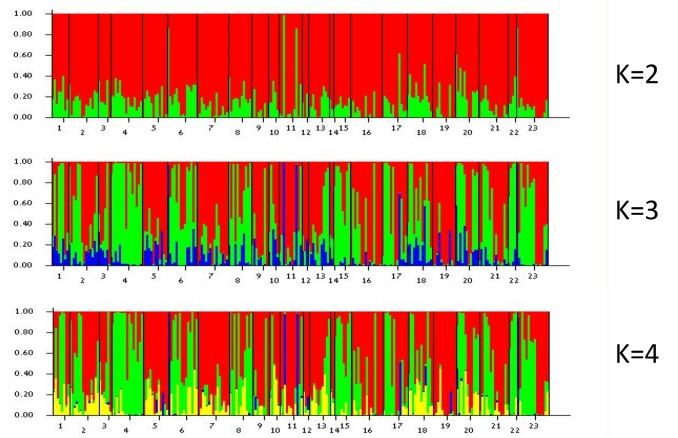
Plot-bar of STRUCTURE Bayesian clustering of multilocus *A. coronaria* genotypes (462 polymorphic loci). K = a putative number of populations ranging from K = 1 to K = 23. Numbers under the bar plot are as following: 1. Mt. Meron—red, 2. Mt. Meron—white, 3. Mt. Meron – purple-blue, 4. Zippori—red, 5. Zippori—white, 6. Zippori—purple-blue, 7. Haifa University—red, 8. Nof Carmel—red, 9. Nof Carmel—purple-blue, 10. Alonim—red, 11. Alonim—white, 12. Alonim—purple-blue, 13. Beit Shearim—red, 14. Beit Shearim—white, 15. Balfouria—red, 16. Balfouria—white, 17. Balfouria—purple-blue, 18. Megiddo—red, 19. Megiddo—white, 20. Megiddo—purple-blue, 21. Purra—red, 22. Purra—red-peach, 23. Lahav—red.

**Figure 5 plants-09-00397-f005:**
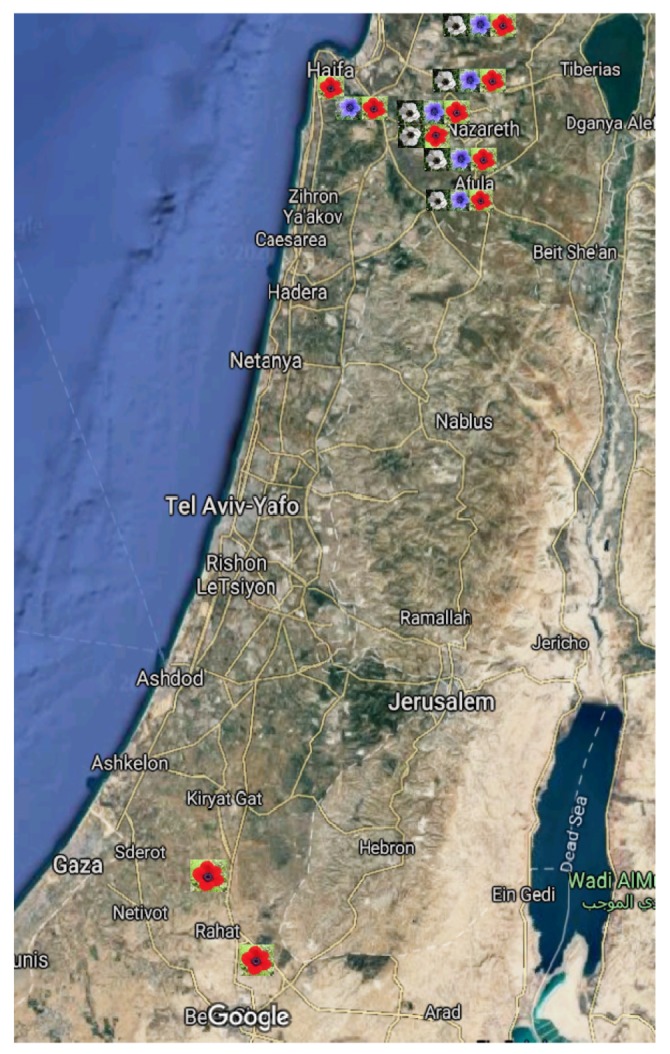
Google map of Israel with indication of our study sites and their composition of floral colour morphs.

**Table 1 plants-09-00397-t001:** The number of landings of various insects on different colour morph flowers of *A. coronaria*, in 2014; the total number of flowers of each colour morph (pooled data from four populations in three observation sites. P value of Spearman’s χ^2^ test with df = 2.

Visitor	White Flowers	Purple-Blue Flowers	Red Flowers	Total	*P*
Honeybee	520	252	0	772	<0.005
*Andrena* sp.	75	16	0	91	<0.005
Syrphid flies	24	7	0	31	0.056
Beetles	0	0	1	1	
Total visits	619	275	1	895	
Number of flowers	694	377	115	1015	

**Table 2 plants-09-00397-t002:** The number of sampled plants (N), its average gene diversity (He—unbiased expected heterozygosity) and SE, and percent of polymorphism (*P*%) in each population (All), and each colour morph (Red, White, Purple-blue), as revealed by AFLP analysis (462 polymorphic loci).

	All				Red			White			Purple-Blue		
Population	*N*	*He*	*SE*	*P*(%)	*N*	*He*	*SE*	*N*	*He*	*SE*	*N*	*He*	*SE*
Mt. Meron	28	0.156	0.007	87.7	14	0.139	0.007	6	0.133	0.008	8	0.191	0.008
Zippori	41	0.141	0.007	89.2	12	0.109	0.006	14	0.153	0.007	15	0.114	0.007
Haifa University	15	0.115	0.006	66.9	15	0.115	0.006	--	--	--	--	--	--
Nof Carmel	19	0.136	0.007	77.9	8	0.096	0.007	--	--	--	11	0.153	0.007
Alonim	19	0.120	0.006	67.8	11	0.096	0.007	3	0.109	0.009	5	0.154	0.008
Beit Shearim	12	0.14	0.007	63.6	10	0.14	0.007	2	0.097	0.01	--	--	--
Balfouria	35	0.094	0.006	60.6	15	0.074	0.005	12	0.094	0.007	8	0.101	0.008
Megiddo	34	0.150	0.006	93.7	11	0.077	0.006	11	0.177	0.008	12	0.178	0.007
Purra	18	0.108	0.007	60.0	14	0.097	0.007	--	--	--	--	--	--
Lahav	15	0.084	0.007	39.4	14	0.085	0.007	--	--	--	--	--	--
Total	236	0.124	0.002	70.7	124	0.102	0.006	48	0.136	0.007	59	0.146	0.007

**Table 3 plants-09-00397-t003:** Study sites, their GPS (Global Positioning System), floral colour morphs (R-red, B-purple-blue, and W-white), and the analyses performed, visitor activity in 2011–12 and 2014, plant material collected for DNA analyses and colour analyses.

Site Name	GPS N E	Colour Morphs	Visitor Activity 2011-12	Visitor Activity 2014	DNA Analyses	Colour Analysis
1. Alonei Abba	32.73	RBW		✔		
	35.17					
2. Alonim	32.72	RBW	✔		✔	
	35.15					
3. Balfuria	32.63	RBW			✔	
	35.29					
4. Beit Lehem Haglilit*	32.74	RBW		✔		
	35.19					
5. Beit Shearim	32.70	RW			✔	
	35.13					
6. Haifa University	32.75	R			✔	
	35.03					
7. Lehavim	32.37	R			✔	
	34.85					
8. Megiddo	32.59	RBW	✔	✔	✔	✔
	35.23					
9. Mt. Meiron	33.01	RBW			✔	
	35.40					
10. Nahal Sanin	32.64	RBW	✔			
	35.10					
11. Nof Carmel	32.75	RBW			✔	
	35.05					
12. Pura	31.50	R			✔	
	34.78					
13. Tsippori	32.76	RBW	✔		✔	
	35.24					

* Two populations.

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
