# Peer review of "Flower Colour Polymorphism, Pollination Modes, Breeding System and Gene Flow in Anemone coronaria"

_plants, 2020, doi:10.3390/plants9030397_

Round 1

Reviewer 1 Report

This manuscript aims to explore the dependence of colour polymorphism of the species Anemone coronaria on its pollination modes, particularly the involvement of different insect guilds, as this is related to the genetic structure of its populations. This paper is an addition to the knowledge assembled through years by several researchers in Israel, including the authors of this ms, on the colour-related pollination ecology of Anemone coronaria in that country.

The study adds to what we know mainly by documenting the importance of wind-pollination of the species Anemone coronaria, which I find is the most important finding of this work together with the physiological perception of flower colour morphs by honeybee. The findings on the genetic similarity (?) among population and flower morphs and the high genetic diversity within each population (thus no differentiation among populations or between colour morphs), does not really clarify colour polymorphism in relation to pollinator preference.

An important issue that is missing, according to my knowledge, from this ms is self-compatibility if the species, and especially within each morph. I do wonder why there has not been even a test applied to check whether the three morphs are self-compatible (so to check the possibility of colour maintenance in a population); hand-cross pollinations between different morphs would also add to our understanding.

Despite the above shortcomings, the ms has merits. For this reason my proposal is that the authors should be given the opportunity to profoundly revise the text and come with a sounder manuscript. To this direction I hope my comments (above and below) will be helpful.

Specific cooments

lines 109-112, 322-5: the carpels of Anemone are uniovulate, so in both these ms places there is a confusion between number of carpels (as they are referred) and numbers of flowers (which probably is the case).

Lines 116 vs. 119 and others: is it seed set of fruit set the authors have dealt with?

Line 120, 123 a.o.: free pollination=open pollination; use one term, it is enough.

Lines 126-8: pooling data does not allow disentangle differences among colour morphs (which is the main purpose of this study); I guess the sample size per colour morph used in the study is too small to bring up any differences between open and wind pollination within each morph; this is important, though, given that wind maybe the most important pollination agent during winter time, and the diachronic existence of colour morphs may greatly depend on it. Thus I believe this is what is relevant to the ms and this should only be mentioned here.

Lines 131-2: no reference in the paper on how stigma surface was measured. The same with stigmatic receptivity.

Line 133: 0.05 x 0.491 x 84 = 2.06, not 2.2.

Line 141 and further down, e.g. lines 249-250: could the absence of beetles be related to the period the study was carried out, i.e. Jan-Feb (see lines 249), whereas normally beetles are active at a later time? There are several inconsistencies, in this sense, especially if one reads also other papers on the subject in the same country. For instance, in lines 249-250 it reads that red flowering morph is abundantly flowering between Jan-Feb, which is only partly supported by the reference [54= Dafni et al. 1990] according to which observations were carried out between February 1 to May 30, with red anemones being flowering for 60 days. According to another paper (Keasar et al. 2010) Glaphyridae activity started in mid-February. Yet, in a third paper (Dafni & Potts 2004), Anemone flowers between February–March, and Glaphyrids were observed during March and April. Since there is much year-to-year variation both in flowering times and in insect activity period it would make sense to know whether there was enough overlapping period between red Anemone in flower and Glaphyrid beetles active.

Lines 157-9, Table 1, and several other places: Although GPS co-ordinates are shown in Table 1, this does not help the reader understand about the geographical location of the sites/populations. A map would be very helpful, and inclusion of the relative location (N … S) of the population in Tables (e.g. Table 3) would increase the readability and understanding of the text.

Line 164 and elsewhere: what is K?

Line 189: replace “self-pollination” by “spontaneous self-pollination”. I also find that an important issue missing from this ms is: self-compatibility within each morph (see abve).

Line 216: Not clear what is meant by “high genetic identity among populations (Table 4).” Maybe “similarity” should be used instead of identity? Moreover, there is no Table 4 in the ms, probably Table 3 is meant instead.

Lines 238-242: Do the authors have any explanation for the non-preference of honeybees and Andrena bees to visit red flowers, in contrast to earlier findings? Is it possible that honeybees do not see red because their perceptions of the red flowers lie near the achromatic centre (Fig. 2)? Could this be related to the second flower colour, i.e. the extend of the white ring between the black centre and the red petals?

Line 335: how many pollen traps?

Lines 352-364: no numbers about the individually followed insects are given and how many sessions were carried out; no numbers of different colour morphs given. A systematic separation between years, dates and forms would be necessary.

Tables are not numbered according to their appearance in the text (text starts with Table 2, Table 1 is at the end).

Literature

Dafni, A., Bernhardt, P., Shmida, A., Ivri, Y., Greenbaum, S., O’Toole, C.H., Losito, L. Red Bowl-Shaped Flowers: Convergence for Beetle Pollination in the Mediterranean Region (1990) Israel Journal of Botany, 39 (1-2), pp. 81-92.

Dafni, A., Potts, S.G. The role of flower inclination, depth, and height in the preferences of a pollinating beetle (Coleoptera: Glaphyridae)(2004) Journal of Insect Behavior, 17 (6), pp. 823-834.

Keasar, T., Harari, A.R., Sabatinelli, G., Keith, D., Dafni, A., Shavit, O., Zylbertal, A., Shmida, A.Red anemone guild flowers as focal places for mating and feeding by Levant glaphyrid beetles(2010) Biological Journal of the Linnean Society, 99 (4), pp. 808-817.

Author Response

Responses to the comments of Rev 1

Our responses follow the original comments in bold italics letters.

The findings on the genetic similarity (?) among population and flower morphs and the high genetic diversity within each population (thus no differentiation among populations or between colour morphs), does not really clarify colour polymorphism in relation to pollinator preference.

However, we think that these results are important for determining whether actual gene flow occurs mainly within each colour morph due to pollinators’ activity, which may lead to intraspecific divergence, or it occurs homogenousely also among all colour morphs and populations as our results indicate. 

An important issue that is missing, according to my knowledge, from this ms is self-compatibility if the species, and especially within each morph. I do wonder why there has not been even a test applied to check whether the three morphs are self-compatible (so to check the possibility of colour maintenance in a population); hand-cross pollinations between different morphs would also add to our understanding.

Concerning self incompatability and crossing between flower colour morphs in the discussion we added a citation of a recent research that gave the answers:” Field experiments proved that A. coronaria was strongly self-incompatible. Hybridisation experiments showed that the different color morphs of A. coronaria could be readily crossed as they produced fruits with viable seeds. Saoud, N. S. (2005). Biosystematics of Anemone coronaria L. and related species (Doctoral dissertation, University of Reading).

Horowitz (1995)  (ref # 51) already tested it: “progeny tests of control crosses have led to the identification four major loci that determine perianth colour epistatically”.  Later in the paper she described heterozygotes of the crosses. 

Despite the above shortcomings, the ms has merits. For this reason my proposal is that the authors should be given the opportunity to profoundly revise the text and come with a sounder manuscript. To this direction I hope my comments (above and below) will be helpful.

Specific comments:

lines 109-112, 322-5: the carpels of Anemone are uniovulate, so in both these ms places there is a confusion between number of carpels (as they are referred) and numbers of flowers (which probably is the case). In both places we actually counted the number of carpels per flower as written in the text.

Lines 116 vs. 119 and others: is it seed set of fruit set the authors have dealt with? We have changed “seed set” to “fruit set”.

Line 120, 123 a.o.: free pollination=open pollination; use one term, it is enough. We have changed “open-free pollination” to “free pollination”.

Lines 126-8: pooling data does not allow disentangle differences among colour morphs (which is the main purpose of this study); I guess the sample size per colour morph used in the study is too small to bring up any differences between open and wind pollination within each morph; this is important, though, given that wind maybe the most important pollination agent during winter time, and the diachronic existence of colour morphs may greatly depend on it. Thus I believe this is what is relevant to the ms and this should only be mentioned here. The reviewer is correct; therefore we don’t present the comparison between open and net covered flowers for each colour morph separately. Here we present only the results, and their importance is presented in the discussion.

Lines 131-2: no reference in the paper on how stigma surface was measured. The same with stigmatic receptivity. We added:”( measured under microscope)”.

Line 133: 0.05 x 0.491 x 84 = 2.06, not 2.2. Corrected to 2.1

Line 141 and further down, e.g. lines 249-250: could the absence of beetles be related to the period the study was carried out, i.e. Jan-Feb (see lines 249), whereas normally beetles are active at a later time? There are several inconsistencies, in this sense, especially if one reads also other papers on the subject in the same country. For instance, in lines 249-250 it reads that red flowering morph is abundantly flowering between Jan-Feb, which is only partly supported by the reference [54= Dafni et al. 1990] according to which observations were carried out between February 1 to May 30, with red anemones being flowering for 60 days. According to another paper (Keasar et al. 2010) Glaphyridae activity started in mid-February. Yet, in a third paper (Dafni & Potts 2004), Anemone flowers between February–March, and Glaphyrids were observed during March and April. Since there is much year-to-year variation both in flowering times and in insect activity period it would make sense to know whether there was enough overlapping period between red Anemone in flower and Glaphyrid beetles active.

The reviewer is correct in pointing to the huge differences in flower visitors’ activity in different years in our study as well in comparison to other studies performed earlier in Israel. Therefore, in the discussion at the end of the “Flower visitors” section we added: “The differences in the composition and frequency of flower visitors to A. coronaria flowers, in our study and among other studies [54, 55, 56], may reflect the inter annual high variability in temperatures and precipitation amounts during the Mediterranean winter.”.

 Lines 157-9, Table 1, and several other places: Although GPS co-ordinates are shown in Table 1, this does not help the reader understand about the geographical location of the sites/populations. A map would be very helpful, and inclusion of the relative location (N … S) of the population in Tables (e.g. Table 3) would increase the readability and understanding of the text. A map was added in Figure 5.

Line 164 and elsewhere: what is K? This is explained in the Methods : The inference of the probable number of clusters was extracted by the log likelihood for each putative number of populations (K), Ln P(D) = L(K), and by the delta K method [64], using the program Structure Harvester [65].

Line 189: replace “self-pollination” by “spontaneous self-pollination”. Corrected.

I also find that an important issue missing from this ms is: self-compatibility within each morph (see abve).

See our response above.

Line 216: Not clear what is meant by “high genetic identity among populations (Table 4).” Maybe “similarity” should be used instead of identity? This is a well-known genetic index and presented as Nei genetic identity when presenting the results.

 Moreover, there is no Table 4 in the ms, probably Table 3 is meant instead. (Table 4) was deleted.

Lines 238-242: Do the authors have any explanation for the non-preference of honeybees and Andrena bees to visit red flowers, in contrast to earlier findings? Is it possible that honeybees do not see red because their perceptions of the red flowers lie near the achromatic centre (Fig. 2)? Could this be related to the second flower colour, i.e. the extend of the white ring between the black centre and the red petals?

We have added: ”The differences in honeybees visitations may reflect differences in the environmental conditions as well as in the co-flowering species in the different studies.”

Line 335: how many pollen traps? We have added: ”we constructed 30 pollen traps”.

Lines 352-364: no numbers about the individually followed insects are given and how many sessions were carried out; no numbers of different colour morphs given. A systematic separation between years, dates and forms would be necessary. In the “results” (lines 145-147 )we added: “Four independent observation sessions were made on 6.02.2014 (two sessions), 9.02.2014 and 21.02.2014) in which the sequence of the visits of 13, 120, 30 and 43 individual bees to the various colour morphs were recorded (Table 2).”

Tables are not numbered according to their appearance in the text (text starts with Table 2, Table 1 is at the end). Tables’ numbers were corrected.

Reviewer 2 Report

Review of 'Flower colour polymorphism, pollination modes, breeding system and gene flow in Anemone coronaria' by Dafni et al. submitted to Plants, February 2020

In this excellent manuscript, Dafni et al. describe the pollination ecology of Anemone coronaria populations along a rainfall gradient extending from the north to the south of Israel. The three color morphs they studied vary in petal reflectance profile and tendency to draw specific types of pollinators, but surprisingly for such showy flowers, they also rely on wind-dispersed pollen for assuring fertilization. The authors show that there is no significant genetic differentiation among the different populations, and the random gene flow enforced by wind pollination is the likely explanation for the relative genetic uniformity. Intriguingly, red-monomorph populations prevail in the south, and the authors suggest the red form could be an adaptation to herbivory by cattle and/or be linked to genes that confer resistance to aridity.
This paper is well-written and very interesting. The authors have deployed a range of techniques, from reflectance measurements to AFLP analysis, and the statistical support for their conclusions is solid.
A few sentences are tricky to understand because of awkward punctuation: parentheses in line 136; semi-colons in line 138 (here, it might be best to set off the numbered bullets by enclosing the numbers in brackets). In line 138, 'expected' might work better than 'supposed'. In line 335, I think they mean 'circles' or 'patches', not 'holes'; if you had holes in the plastic strap, there'd be nothing for the sticky material to adhere to. In line 407, a better construction would be to write about the colors 'as perceived by humans' and 'sensed by pollinators…', because the colors themselves aren't human.
Regarding line 204, I would not expect wind pollination to be effective under rainy conditions. It seems to me that ambophily in this species could be a mechanism for maintaining its generalist pollination syndrome, which would shelter A. coronaria from pollinator-driven selection for more narrowly targeted floral characteristics and the inherent risks of multiple small subpopulations each with a reduced pollinator base. That would also explain why you have just the one species in Israel.
In any case, the authors have demonstrated convincingly that A. coronaria is a rich system from pollination ecology studies and I look forward to answers in the future regarding some of the lingering mysteries highlighted in this paper.

Author Response

Responses to the comments of Rev 2

Our responses follow the original comments in bold italics letters.

A few sentences are tricky to understand because of awkward punctuation: parentheses in line 136; semi-colons in line 138 (here, it might be best to set off the numbered bullets by enclosing the numbers in brackets). Corrected

In line 138, 'expected' might work better than 'supposed'. Corrected.

In line 335, I think they mean 'circles' or 'patches', not 'holes'; if you had holes in the plastic strap, there'd be nothing for the sticky material to adhere to. Corrected to: “pollen traps from plastic straps with 28.27 mm2 holes and sticky tape underneath them”.

In line 407, a better construction would be to write about the colors 'as perceived by humans' and 'sensed by pollinators…', because the colors themselves aren't human. Corrected as suggested.
Regarding line 204, I would not expect wind pollination to be effective under rainy conditions. We added the following explanation: “Anemone coronaria flowers close at night and during rainy days, thus its pollen is protected and wind pollination can occur during sunny days that are not uncommon in the mild Mediterranean winter.”.

It seems to me that ambophily in this species could be a mechanism for maintaining its generalist pollination syndrome, which would shelter A. coronaria from pollinator-driven selection for more narrowly targeted floral characteristics and the inherent risks of multiple small subpopulations each with a reduced pollinator base. That would also explain why you have just the one species in Israel. This seems to us as a highly speculative explanation.

Reviewer 3 Report

  1. Introduction contains ten paragraphs without and apparent link among most of them.
  2. L 33-36. Although some authors have considered the study of “continuous” characters as “polymorphism”, this is an extended use of the definition of “polymorphism” (see Ford 1945, Biological Review, 20, 73–88). The same is true for the application of polymorphism within or between populations.
  3. It is difficult two follow the distribution of color morph in Israel. I recommend making a figure with the approximate distribution of color morph that also may contain the study sites of table 1.
  4. Figure 2. What means the circle and the line inside the hexagon? Authors need to explain what means that red points are located inside the circle.
  5. L 119-128. I’m not clear why authors perform two different K-W analyses: one for color morphs and other for pollination treatment. Why not an ANOVA or GLM with two factor and interaction?
  6. L 122. I don’t understand what mean the reference 55 here. It is means that the data of stigma receptivity is from this reference?
  7. In the wind pollination treatment they counted the number of flowers setting fruits but the number of achenes was not counted. It means that a flower with only one achene (with more than 500 ovules) was considered a transformed fruit? In this case, the response is binary (0 or 1) and K-W analysis is not the best option.
  8. L 145-149 and Table 2. To establish pollinator preferences is one of the main goals of the paper, but I not clear about the results and conclusions of the study. Data of the table 2 are first landing or total landings? This is critical because if the authors “followed individual visitors” (L 357), they are measuring flower constancy (Waser 1986, American Naturalist 127: 593-603) and the analysis will be different that they performed in table 2. It would be interesting to the author show the flower color frequency.
  9. L 181-184. The model of vision for Eristalis tenax (Syrphidae) is well established and can be taking into account for A. coronaria spectral data (Troje 1993, Naturforschg 48: 96–104; Lunau 2014, J Comp Physiol A).
  10. L 329-321. Authors indicate the threshold level for color loci differentiation in the color hexagon but they don’t apply to their data. It will be interesting to know these values for A. coronaria.
  11. I think that in studies of color polymorphism is important for the reader to see the color of flowers, even in electronic journals, which has less space restrictions. Thus, I strongly recommend to the authors to include photography of each color morphs, or even a photo of a polymorphic population. They can build a composed figure including photos of color morphs and their reflectance (currently Fig. 1).

Author Response

Responses to the comments of Rev 3

Our responses follow the original comments in bold italics letters.

  1. Introduction contains ten paragraphs without and apparent link among most of them. We think that the introduction is well structured. It begins with introducing the phenomenon of floral intraspecific colour polymorphism that in case of pollinator constancy may cause divergent evolution and thus is interesting. Is continues with the various mechanisms that maintain and might explain floral polymorphism. The introduction ends with summarizing our current knowledge on A. coronaria in Israel and finally presents the aims of our research.

  1. L 33-36. Although some authors have considered the study of “continuous” characters as “polymorphism”, this is an extended use of the definition of “polymorphism” (see Ford 1945, Biological Review, 20, 73–88). The same is true for the application of polymorphism within or between populations. We changed to: “FCP may include cases of gradual [1], but mainly refers to discrete differences in flower colours among morphs [2-6].”

  1. It is difficult two follow the distribution of color morph in Israel. I recommend making a figure with the approximate distribution of color morph that also may contain the study sites of table 1. We added a map Figure 5.
  2. Figure 2. What means the circle and the line inside the hexagon? Authors need to explain what means that red points are located inside the circle. The center of the hexagon represents the background colour, and the short distance of the red flowers from it indicates that bees can differentiate them from the background less than white or purple blue flowers.

  1. L 119-128. I’m not clear why authors perform two different K-W analyses: one for color morphs and other for pollination treatment. Why not an ANOVA or GLM with two factor and interaction? As clearly stated, we compared the percentage of flowers that set fruits in each of the treated and not treated pollination cages in each colour morph. Such type of data does not permit the suggested tests.

  1. L 122. I don’t understand what mean the reference 55 here. It is means that the data of stigma receptivity is from this reference?In line 132, ref 55 related for the “average duration of stigmatic receptivity of a flower is 3.5 day (= 84 hours) [55]”.

  1. In the wind pollination treatment they counted the number of flowers setting fruits but the number of achenes was not counted. It means that a flower with only one achene (with more than 500 ovules) was considered a transformed fruit? In this case, the response is binary (0 or 1) and K-W analysis is not the best option. As clearly stated, we compared the percentage of flowers that set fruits in each of the treated and not treated pollination cages in each colour morph. Thus, the data is not binary.

  1. L 145-149 and Table 2. To establish pollinator preferences is one of the main goals of the paper, but I not clear about the results and conclusions of the study. Data of the table 2 are first landing or total landings? This is critical because if the authors “followed individual visitors” (L 357), they are measuring flower constancy (Waser 1986, American Naturalist 127: 593-603) and the analysis will be different that they performed in table 2. It would be interesting to the author show the flower color frequency. Flower colour frequency is presented in Table 1.

  1. L 181-184. The model of vision for Eristalis tenax(Syrphidae) is well established and can be taking into account for A. coronaria spectral data (Troje 1993, Naturforschg 48: 96–104; Lunau 2014, J Comp Physiol A). We have used this model in an earlier version of this MS and were strictly requested to remove it.

  1. L 329-321. Authors indicate the threshold level for color loci differentiation in the color hexagon but they don’t apply to their data. It will be interesting to know these values for coronaria.

We agree with the referee and added the relevant values in the results: “The average (± stdv) location of the red flowers (n=20) was X=-0.08850±0.08869 and Y=-0.03450±0.01050, that of the white flowers (n=20) was X=0.08450±0.03069 and Y=0.14350±0.02207, and that of the purple-blue flowers (n=20) was X= -0.1730±0.04438 and Y=0.1675±0.03024. The pair wise average Euclidan distances between the average locations of the different colour morphs were:
purple-blue - red: 0.261±0.050, purple-blue -white: 0.287±0.050, red-white: 0.269±0.079”.

In the discussion we added: “The threshold values of Euclidean distance between any two colours in the hexagon units above which bees can perceive them as different signals ranges between 0.062 [89, 91] and 0.100 [92]. The distances between the three A. coronaria colour morphs ranged between 0.26 and 0.27. Therefore, it can be assumed that honeybees, and probably also solitary bees, can differentiate among them. The centre of the hexagon represents the background colour, thus shorter distance of the red flowers from this point indicates that bees can differentiate them from the background less than white or purple blue flowers”.

I think that in studies of color polymorphism is important for the reader to see the color of flowers, even in electronic journals, which has less space restrictions. Thus, I strongly recommend to the authors to include photography of each color morphs, or even a photo of a polymorphic population. They can build a composed figure including photos of color morphs and their reflectance (currently Fig. 1). We added  Figure 1, that presents pictures of single flowers as well as of mixed coloured populations.

Round 2

Reviewer 3 Report

The manuscript has been substantially improved after the revision. New figures add valuable information and the incorporation of the Euclidean distances between pairs of color loci in the honeybee hexagon and their interpretation seems appropriate. Only one little commentary (line 308-309), Narbona et al. 2018 includes Anemone coronaria as polymorphic species (Table 1 in pp 10, and Supplementary Table S1).